# GI inflammation Increases Sodium-Glucose Cotransporter Sglt1

**DOI:** 10.3390/ijms20102537

**Published:** 2019-05-23

**Authors:** Jiyoung Park, In-Seung Lee, Kang-Hoon Kim, Yumi Kim, Eun-Jin An, Hyeung-Jin Jang

**Affiliations:** 1College of Korean Medicine, Kyung Hee University, 26, Kyungheedae-ro, Dongdaemun-gu, Seoul 02447, Korea; sdu1771@naver.com (J.P.); leejohn1031@khu.ac.kr (I.-S.L.); poklmoo@naver.com (K.-H.K.); yumi0201@khu.ac.kr (Y.K.); aej3866@naver.com (E.-J.A.); 2Department of Science in Korean Medicine, Graduate School, Kyung Hee University, Seoul 02447, Korea

**Keywords:** GI inflammation, Sglt1, glucose sensing receptor, incretin secretion, GIP, GLP-1

## Abstract

A correlation between gastrointestinal (GI) inflammation and gut hormones has reported that inflammatory stimuli including bacterial endotoxins, lipopolysaccharides (LPS), TNFα, IL-1β, and IL-6 induces high levels of incretin hormone leading to glucose dysregulation. Although incretin hormones are immediately secreted in response to environmental stimuli, such as nutrients, cytokines, and LPS, but studies of glucose-induced incretin secretion in an inflamed state are limited. We hypothesized that GI inflammatory conditions induce over-stimulated incretin secretion via an increase of glucose-sensing receptors. To confirm our hypothesis, we observed the alteration of glucose-induced incretin secretion and glucose-sensing receptors in a GI inflammatory mouse model, and we treated a conditioned media (Mϕ 30%) containing inflammatory cytokines in intestinal epithelium cells and enteroendocrine L-like NCI-H716 cells. In GI-inflamed mice, we observed that over-stimulated incretin secretion and insulin release in response to glucose and sodium glucose cotransporter (Sglt1) was increased. Incubation with Mϕ 30% increases Sglt1 and induces glucose-induced GLP-1 secretion with increasing intracellular calcium influx. Phloridzin, an sglt1 inhibitor, inhibits glucose-induced GLP-1 secretion, ERK activation, and calcium influx. These findings suggest that the abnormalities of incretin secretion leading to metabolic disturbances in GI inflammatory disease by an increase of Sglt1.

## 1. Introduction

The gastrointestinal (GI) tract is an important organ system for digesting food, absorbing nutrients, excreting waste, and hormone secretion. Enteroendocrine cells (EECs), which are specialized cells of the gastrointestinal tract and pancreas, account for 1% of all intestinal cells. More than 30 peptides by different types of EECs have been found to play an important role in the regulation of appetite, glucose homeostasis, lipid metabolism, energy expenditure, and various metabolic functions in response to nutrients or neural signaling (Wu, et al. [1,2,3]). Incretin hormones, known as glucose-dependent insulinotropic polypeptide (GIP) and glucagon-like peptide-1 (GLP-1), are secreted in enteroendocrine K-cells and L-cells, respectively. These hormones play an important role in the regulation of glucose metabolism and homeostasis by insulin secretion in glucose-dependent manner and stimulate the proliferation of pancreatic β-cells [4,5,6,7]. Incretin hormone is immediately secreted by neurotransmitters and luminal nutrients including carbohydrates, proteins, fat derivatives, and non-nutrient stimuli, such as ghrelin, leptin, bile acid, bitter tastes, endotoxins, and cytokines, through diverse sensing receptors [8,9,10,11,12,13,14,15,16]. Among the various environmental stimuli, glucose is the most important energy source and potent stimulator of incretin secretion in the intestine. The mechanism of glucose-induced incretin secretion has been identified as occurring through two alternative pathways: endocrine cells detect glucose by modified taste cells via sodium glucose cotransporter (SGLT1) playing a major role in glucose uptake with the facilitative glucose transporter GLUT2. These transporters induce glucose absorption from the intestinal lumen and liberate hormone granules resulting in electrical activity via closure of ATP-sensitive K^+^ channels (K_ATP_ channels) and the opening of voltage-dependent Ca^+^ channels. As a second pathway, the sweet taste receptor (STR; heterodimer of T1R2 and T1R3) signaling pathway, and they stimulate hormone secretion via Ca^2+^ and the cAMP signaling pathway.

GI disease may be affected by infectious, autoimmune, and physiological state. Inflammatory bowel disease (IBD) is the most well-known GI disease accompanied by inappropriate immunological activation caused by multifactorial etiology. IBD is accompanied by idiopathic inflammatory conditions in the epithelial barrier between the small intestine and the rectum, there are two major types of IBD: ulcerative colitis (UC) and Crohn’s disease (CD). UC and CD are most common in the terminal ileum to colon and have present with common symptoms, such as weight loss, diarrhea, rectal bleeding, and abdominal pain caused by intestinal damage [17,18,19,20]. GI inflammation induces abnormalities in enteroendocrine hormones, and intestinal endotoxins and inflammatory contents flow into the underlying tissue due to increased intestinal permeability after gut barrier injury with disturbances of the gut microbiota. Many studies of alterations of hormone levels have been conducted in cases of GI inflammation, such as in patients with gastritis, colitis, ileitis, irritable bowel syndrome (IBS), and IBD [21,22,23]. In particular, incretin hormone is closely related to the regulation of glucose homeostasis in glucose-dependent manner. Several studies have been reported that exposure to inflammatory stimuli, such as bacterial endotoxins, lipopolysaccharides (LPS), TNFα, IL-1β, and IL-6, induce abnormalities in incretin secretion [24,25]. In intestinal endotoxic conditions, over-secretion of incretin were found to induce insulin secretion and glucose lowering leading to hyperinsulinemia and blood glucose lowering. Many studies have reported changes in intestinal hormone levels in IBD patients and in animal models, but knowledge of how glucose-induced incretin secretion is affected by GI inflammation is still insufficient. We aimed to confirm how GI inflammation affects glucose-induced incretin secretion and to investigate the cause of change in incretin secretion. To confirm a correlation between GI inflammation and alteration of incretin secretion in response to glucose, we conducted GI inflammatory conditions using dextran sodium sulfate (DSS) and inflammatory cytokines. The DSS-induced mouse model is most widely used as a representative human IBD model to investigate the pathophysiological mechanisms and in pharmacological preclinical studies [26,27,28]. 

In the present study, we investigated over-stimulated incretin secretion in response to glucose exposure leading to insulin secretion and lowering blood glucose levels in GI-inflamed mice and investigated that GI inflammatory condition may increase Sglt1. These findings may demonstrate the abnormal incretin secretion leading to metabolic disturbances in GI inflammatory conditions. 

## 2. Results

### 2.1. Incretin Secretion Increased by Glucose Gavage in Gastrointestinal (GI)-Inflamed Mice 

GI inflammatory conditions were induced by DSS administration into acute and chronic groups (Appendix A). Glucose-stimulated GIP secretion showed sharp increases following 10% glucose gavage, especially in the acutely GI-inflamed group (Figure 1A,B). GLP-1 secretion was also increased compared to that in the control group after glucose consumption (Figure 1C), but there was no significant difference in comparison to the chronic group (Figure 1D). Acutely GI-inflamed mice showed hyperinsulinemia in a fasting state, and insulin secretion was more elevated than in the control group following glucose administration (Figure 1E). In contrast, the chronic group showed a similar pattern of insulin secretion without hyperinsulinemia (Figure 1F). The in vivo reactivity to glucose, which is deeply related to insulin secretion, was examined using the oral-glucose tolerance test (OGTT). Both GI-inflamed mouse models showed elevated glucose disposal. After starvation for 16 h, the acute group showed hypoglycemia and the blood glucose levels rose slowly over a long period of time (Figure 1G), whereas the rise in the blood glucose level exhibited a trend similar to that in the control group, and the glucose level was lower in the chronic GI-inflamed mice (Figure 1H). 

### 2.2. Gastrointestinal (GI) Inflammatory Conditions Stimulate Glucose-Induced Incretin Secretion

The GLP-1 and GIP secretion in response to glucose gavage were increased leading to blood glucose lowering in GI-inflamed mice compared to those in the control mice. Many studies have reported that inflammatory cytokines are well known elevating incretin secretion. To confirm whether the increase of incretin levels caused by inflammatory stimuli, we observed incretin secretion in the small intestine in absence of inflammatory stimuli. Medium containing 10% glucose and a DPP4 inhibitor was treated in minced duodenum and ileum tissues which had been washed off to remove inflammatory stimuli. GIP, which is mainly distributed in the duodenum and jejunum, also showed sharply enhanced hormone secretion after 10% glucose exposure (Figure 2A), and the GLP-1 secretion also over-stimulated in the acutely GI-inflamed group compared to the GI-intact group (Figure 2B). When absence of inflammatory stimuli in the chronic group, the levels of incretin were similar to those of in the control group. Glucose-induced GIP secretion (Figure 2C), and GLP-1 secretion did not observe over-secretion in ileum tissue of chronic group (Figure 2D). In the acute group, the over-secretion of incretin by glucose exposure is observed in both duodenum and ileum tissue, whereas there is significant over-secretion of GIP in duodenum tissue of the chronic group. Furthermore, we observed the distribution of chromogranin A (CgA), which is located in the secretory vesicles of endocrine cells. In the small intestine of the acutely GI-inflamed group, CgA-positive cells did not show differences when compared to those of the control group (Figure 2E). CgA-positive spots are (4.3 ± 0.75) in the control group and acute group (4 ± 0.82) per μm in duodenum tissue, and (4 ± 0.53), (4.1 ± 1.12) in ileum tissue, whereas CgA-positive spots were increased in the chronic group (6.7 ± 1.11) with increased hyperplasia more than the control group (5.5 ± 2.63) per μm (Figure 2F). Chronic inflammation increases the amount of secretory vesicles in the small intestine, but not in acutely GI-inflamed mice. In the both groups, immunohistochemistry of GIP in duodenum tissues (Figure 2G), and GLP-1 in ileum tissues (Figure 2H). Incretin-positive cells did not show differences in acutely GI-inflamed mice, but incretin-positive spots were increased in chronic GI-inflamed mice. These results signify that GI inflammation increases glucose-stimulated incretin secretion and long-term weak exposure of inflammatory stimuli leads to an increment of secretory vesicles containing GIP and GLP-1. 

### 2.3. Na+-Glucose Co-Transporter Sglt1 Protein is Increased in GI-Inflamed Mice

We confirmed the over-secretion of incretin in response to glucose exposure in GI-inflamed state, in vivo and ex vivo. Based on our results, we hypothesized that GI inflammatory condition induces alteration of glucose-sensing receptors leading to increased incretin secretion. Glucose-induced incretin secretion is stimulated via glucose-sensing receptors such as sweet taste receptor (STR), Sglt1, and GLUT2 in enteroendocrine cells. In duodenum tissue, the STR proteins of T1R2, T1R3, and Gα gustducin were reduced compared to the control group (Figure 3A). The expression levels of STR in both ileum tissues were also diminished, but the amount of Sglt1 protein was increased in only the acutely GI-inflamed mice (Figure 3B). GI Inflammatory conditions caused an increase of Sglt1 protein leading to elevating glucose-induced incretin secretion in the small intestine, especially in severe GI-inflamed mice. 

### 2.4. Inflammatory Condition Induces an Increment of Na+-Glucose Co-Transporter Sglt1

To confirm whether inflammatory conditions increase Sglt1 protein in the small intestine, inflammatory stimuli secreted from LPS-stimulated RAW264.7 cells were treated in isolated intestinal epithelium cells. Following LPS stimulation for 24 h, stimulated RAW264.7 cells released cytokines such as IL-1β, IL-6, and TNFα, but there was no difference in IFNγ level (Figure 4A). Intestinal epithelium cells were observed of single cells and crypts particles. Conditioned media (Mϕ 30%) containing a supernatant at 30% of activated-macrophage did not induce morphological changes and cell proliferation after Mϕ 30% treatment for 48 h. Sglt1 was increased in single cells (Figure 4B) and crypts particles (Figure 4C) with incubation of Mϕ 30% media for two days. There was a significant increase of IF intensity (Figure 4D), and amount of Sglt1 protein was elevated by 186.44% (± 5.48) in intestinal epithelium cells (Figure 4E). In addition to isolated intestinal epithelium cells, a treatment of conditioned media for 48 h in enteroendocrine-like NCI-H716 cells leads to increment of Sglt1 protein in dose-dependent manner (Figure 5A,B). We confirmed increasing glucose-induced incretin secretion and Sglt1 protein, but STRs did not show significant changes. When cultured with conditioned media, GLP-1 secretion was increased by glucose at 110.40% (± 17.05) compared with control, and treatment with the Sglt1 inhibitor phloridzin reduced GLP-1 secretion at 52.55% (± 4.20) in enteroendocrine-like NCI-H716 cells (Figure 5C). Chromogranin A (endorine marker) was seen, but incretin levels were too low (<out of range) in isolated intestinal epithelium cells. ERK phosphorylation also increased in Mϕ 30%-differentiated NCI-H716 cells than control group (Figure 5D), and these ERK activation was reduced by incubation with phloridzin (Figure 5E). GLP-1 is secreted by 10% glucose exposure through an increase of intracellular Ca^2+^ influx in differentiated NCI-H716 cells. Fluo-4 fluorescence, which correlated with intracellular Ca^2+^ entry, is significantly increased after glucose treatment for 1 h and Sglt1 inhibition reduces up-regulation of intracellular Ca^2+^ (Figure 5F). 

## 3. Discussion

In the present study, we aimed to investigate how incretin secretion is affected in the GI inflammatory state. We confirmed elevating incretin secretion in response to glucose exposure through an increase of the Sglt1 in the intestine and enteroendocrine cells. A convoluted connection between the immune and enteroendocrine systems has been reported in various inflammatory diseases, such as infections, obesity, diabetes, and inflammatory bowel disease [29,30], and many studies show alterations of hormone levels in various cases of GI inflammation, including in patients with gastritis, colitis, ileitis, irritable bowel syndrome (IBS), and IBD [21,22,23]. Cross-talk between inflammation and gut hormones has been reported, GI Inflammation impacts on enteroendocrine cells, leading to functional defects and hormone imbalances [31,32,33]. In addition, several clinical studies with IBD patients have reported enteroendocrine disorders and abnormalities of insulin secretion [34,35]. Inflammatory stimuli, such as IL-6, TNFα, and endotoxin, induce GIP and GLP-1 secretion, leading to hyperinsulinemia in critically ill patients [36,37]. In IBD mouse models, portal endotoxin levels and various cytokine levels have been observed at high concentrations with intestinal epithelial inflammation, and plasma GIP and PYY levels were increased [38,39]. Lebrun et al. reported that luminal LPS accelerates GLP-1 secretion through IL-6 and TLR4-dependent manner in GI-inflamed mice [40]. Numerous studies have been reported an increase of GLP-1 secretion in human IBD patients and systemic plasma cytokines impact on intestinal hormone secretion. Many studies have shown that inflammatory cytokines accelerate GLP-1 secretion, while GIP secretion is stimulated with LPS and these secretions are mediated by the IL-1β-dependent pathway, but not IL-6 and TNFα [41]. Inflammatory stimuli are well known to be involved in intestinal hormone release, and we also observed high concentrations of plasma incretins as well as PYY (data not shown) in a GI-inflamed mouse model using DSS administration. 

In our results, extracellular inflammatory conditions, such as cytokines and endotoxin, cause the high levels of incretin hormones have been demonstrated by the absence of inflammatory stimuli. Of note, oral glucose administration stimulates robust incretin secretion in GI-inflamed mice, and increasing glucose-induced incretin secretion leads to lowering blood glucose levels via insulin secretion. Interestingly, DSS administration mainly affects colon but the small intestine is affected by GI inflammation. We confirmed that elevated incretin levels decreased when extracellular inflammatory stimuli were removed, and over-secretion of incretin in response to 10% glucose was observed in the minced small intestine of GI-inflamed mice than those of GI-intact mice, but there was not significant compared with in the intact control group except acutely GI-inflamed duodenum tissue. We confirmed over-secretion of incretin hormone in response to glucose exposure in the acute inflammatory group more than that in the chronic group. Over-secretion of incretin hormone was due to the elevated Sglt1 protein in the acute group, whereas the chronic group was affected by the increase of hormone production rather than the increase of Sglt1 protein. The effect of Sglt1 on glucose-induced incretin secretion is greater than that of hormone production. 

SGLT1 induces membrane polarization by the co-transport of Na^+^ with glucose playing a critical role in glucose-dependent incretin secretion, but impact of Sglt1 on plasma GLP-1 levels in vivo is controversial. Sun et al. demonstrated that Sglt1 plays a role in glucose-induced GLP-1 secretion using sglt1 inhibitor phloridzin in human ileal L-cells. Gorboulev et al. reported that glucose does not stimulate an incretin hormone release in SGLT1-deficient mice [42]. An increase of incretin secretion in response to glucose exposure is caused by elevated incretin production or increased amount of receptor to recognize glucose. In our results, we found that the Sglt1 protein is specifically increased in the small intestine of GI inflammatory conditions, while STR showed lower levels of intestinal protein expression than GI intact mice. GLUT2 did not show significant change compared to those in GI intact mice, although it is well known that GLUT2 does not contribute significantly to incretin secretion in the intestine. 

To confirm whether Sglt1 protein by inflammatory stimuli were increased, isolated intestinal epithelium cells and NCI-H716 cells were incubated with conditioned media (Mϕ 30%) containing IL-1β, IL-6, and TNFα released from LPS-stimulated macrophage cells. Pro-inflammatory cytokines TNFα and IL-6 are associated with many chronic inflammatory diseases. Gagnon et al. reported that chronic exposure of TNFα induces impairment of GLP-1 secretion, and TNFα interrupted GIP-induced GLP-1 secretion through inhibiting GIP-induced AKT phosphorylation in NCI-H716 cells [43]. Whereas, Ellingsgaard et al. reported that IL-6 enhances GLP-1 production from pancreatic alpha cells and mRNA expression associated with GLP-1, such as *gcg, pcsk1*, and *slc5a1* in GLUTag cells [10]. Moreover, GLP-1 secretion resulting from gut ischemic injury is immediately regulated through the IL-6-dependent pathway and GLP-1 secretion by LPS injection was mitigated in gene deletion of *IL-6* (*IL6*
^−/−^) mice. It is well known that the IL-6-depedent pathway plays a prominent role in the correlation between GLP-1 secretion and GI inflammation. We assumed that inflammatory cytokines might increase the expression of Sglt1 protein. Following incubation with conditioned media containing inflammatory cytokines, Sglt1 protein was increased in isolated intestinal epithelium cells and NCI-H716 cells, but not other glucose-sensing receptors STR and enteroendocrine marker chromogranin A (CgA). Although glucose-induced incretin secretion in intestinal epithelium cells was not measured (<out of range), GLP-1 secretion by 10% glucose exposure was elevated in differentiated NCI-H716 cells with inflammatory conditions. The glucose-induced downstream signals, such as ERK phosphorylation and calcium influx, were also increased in NCI-H716 cells exposed to inflammatory stimuli compared to the absence of inflammatory stimuli. Although ERK phosphorylation is known to be associated with GLP-1 secretion, a clear mechanism of ERK activation is still unknown. In our data, ERK phosphorylation is increased by glucose exposure. Calcium influx is increased by ATP production after glucose uptake and signal transduction through various receptors. We used the sglt1 inhibitor phloridzin to confirm whether incretin secretion may induced by an increase of Sglt1, a treatment of phloridzin abolished the Ca^2+^ increase. Moreover, we treated each cytokine to differentiated NCI-H716 cells to find a specific cytokine that induced an increase of Sglt1 in identical manner, but sole treatment of human recombinant cytokine including IL-1β, IL-6, TNFα, IFNγ, and LPS did not show an increase of Sglt1 protein (data did not show). 

Accordingly, we observed that GI inflammatory conditions promotes glucose-induced incretin secretion by an increase of Sglt1 protein. Furthermore, we investigated the hormone secretion and blood glucose levels in interval fasting mice for three days to confirm whether fasting and reduced food intake after DSS administration effect on incretin secretion (Appendix A). There were no changes in hormone secretion and blood glucose levels during fasting. Moreover, lowering contents of glucose and fetal bovine serum did not induce an increment of Sglt1 protein in NCI-H716 cells. These results showed enhanced incretin secretion in response to glucose through an increase of Sglt1 protein induced by GI inflammatory conditions, not fasting. Our results show a link between GI inflammation and glucose-dependent incretin secretion. Severe inflammatory condition leads to more Sglt1 increase than mild inflammation. These findings demonstrate the possibility that GI inflammation causes metabolic disturbances through alterations of glucose-induced incretin secretion. More studies are required to identify mechanism and specific factors that affect receptors which react to various stimuli in inflammatory conditions. Collectively, our data may explain the abnormalities of intestinal hormone secretion are caused by GI inflammatory conditions through increasing Sglt1 expression.

## 4. Material and Methods

### 4.1. Mice and DSS-Induced GI Inflammation 

This study was carried out in strict accordance with the ethical guidelines of Kyung Hee University. All animal study protocols were approved by the Institutional Animal Care and Use Committee (IACUC) of Kyung Hee University (confirmation number: KHUASO-(SE)-14-043). Specific pathogen-free female C57BL/6 mice, 8–9 weeks old, were purchased from Daehan Biolink (DBL, Chungcheongbuk-do, Republic of Korea). All mice were acclimated for a week. DSS (MP biomedicals, Solon, OH), dissolved in autoclaved water to 3% (*w/v*) or 2% was administered ad libitum. The acute group were induced 3% DSS administration for seven days and chronic group were induced by seven days of oral 2% DSS followed by 10 days of H_2_O (for 1, 2, and 3 cycles). Acute and chronic mice were sacrificed after DSS-induction at seven and 52 days, respectively

### 4.2. Assessment of Clinical Score and Histological Score

The mice were monitored and scored daily using standard chart. For body weight, loss of 0–3% weight was scored as 0, a loss of 3–10% weight was registered as 1, a loss of 10–20% weight was registered as 2, loss of >20% weight was registered as 3. For the diarrhea score, zero points were assigned for well-formed pellets, two points for pasty and semi-formed stools that did not adhere to the anus, and three points for watery stools that smeared to the anus. For visible feces blood, feces were squashed and then assessed for the degree of feces blood (Appendix A). Intestinal sample of damaged intestine collected at seven and 52 days from each mouse for histological evaluation and fixed by 10% formalin for 24 h and embedded in paraffin. Paraffin sample were cut into 5 µm sections and stained with H and E [44]. The histological results were scored based on the presented in Appendix A and examined for damage. All scorings were performed in a blinded fashion (Appendix A).

### 4.3. Oral Glucose Tolerance Test and Blood Sampling for Multiplex Assay

The method of oral glucose tolerance test and blood sampling were described in our previous studies [45]. Briefly, mice were starved for 16 h and then administrated with 5 mg/kg glucose after the 0 time point.

### 4.4. Measurement of Gut Hormones and Cytokines in Blood

Mice were fasted for 16 h and sacrificed by CO_2_. Collected blood from abdominal aorta was transferred into EDTA coated-tube containing DPP-4 inhibitor and 4-(2-aminoethyl) benzenesulfonyl fluoride hydrochloride (AEBSF) (Sigma-Aldrich, Saint Louis, MO, USA) for protecting blood coagulation and peptide hormone degradation. The plasma was immediately separated into fresh tubes after centrifugation (1000× *g*, 10 min, 4 °C). The levels of gut hormones and cytokines were measured using Milliplex^®^ MAP Mouse Metabolic Magnetic Bead Panel (EMD Millipore Corporation, Billerica, MA, USA) and Bio-Rad Mouse Cytokine Assay (Bio-Rad, Hercules, CA, USA). The plasma concentration was measured by a Bio-Plex^®^ MAGPIX™ Multipex reader and performed as described in each manufacturer’s guide using Bio-Plex manager software (Bio-Rad).

### 4.5. Immunohistochemistry and Immunofluorescence

Parts of intestine tissue and pancreas tissue were dewaxed, rehydrated, and then dipped in the antigen-retrieval solution (10 mM sodium citrate, pH 6) in microwave oven for 30 min, followed by 0.3% hydrogen peroxide for 30 min, and overnight incubation at 4 °C with primary antibodies. Information of primary antibodies were entered in Appendix A. After primary antibody incubation for IHC, tissues were incubated for 30 min with secondary antibodies, and then incubated slides using a Vectastain ABC Kit (Vector laboratories, Burlingame, CA, USA) for 1 h, followed by DAB staining (Vector) and counterstaining with hematoxylin. For immunofluorescence, we used secondary antibodies including donkey anti-rabbit Alexa Fluor 488, donkey anti-goat Alexa Fluor 594 (Life Technology, Carlsbad, CA), and DAPI. The method of immunofluorescence was described in our previous studies [46]. 

### 4.6. Tissue Culture of Mouse Small Intestine

Mice intestines were removed from stomach to anus and then 5 cm was cut from the GI junction and cecum as each duodenum and ileum pieces, followed by perfused twice ex vivo using cold PBS containing the protease inhibitor cocktail (Roche, Basel, Swiss) and minced [47]. Minced duodenum and ileum tissue were transferred into 24-well plates and immediately incubated with or without 10% glucose Dulbecco’s modified Eagle’s medium (DMEM) containing DPP4 inhibitor (Millipore) for an hour. The incubated supernatants of minced intestinal tissues were collected and centrifuged at 1000× *g* for 20 min at 4 °C for GIP and GLP-1 measurements (Bio-Rad). 

### 4.7. Cell Culture and Preparation of Conditioned Media

The murine macrophage cell line RAW264.7 cells and human cell line NCI-H716 cells were purchased from the Korean Cell Line Bank (Seoul, Republic of Korea). The culture medium of RAW264.7 was high glucose (4.5 g/L) DMEM (Corning, NY, USA) containing 10% FBS and 1% P/S. RAW264.7 cells were seeded at 7 × 10^6^ cell per T75 flask overnight, and then the culture medium was exchanged with lipopolysaccharide (LPS) at 1 mg/mL for 24 h. Following LPS stimulation, activated-RAW264.7 cells washed with PBS twice and cell medium was changed complete DMEM without LPS for 24 h. Cell supernatant was filtered at 0.22 μm and stored at −80 °C until used. Briefly described summary about preparation of conditioned media in Appendix A. The culture medium of suspension NCI-H716 cells was RPMI (Corning, NY, USA) containing 10% FBS and 1% P/S. To differentiate enteroendocrine-like cells, NCI-H716 cells were seeded in matrigel-coated plates and incubated with high glucose (4.5 g/L) DMEM medium containing 10% FBS and 1% P/S at 37 °C under a 5% CO_2_ condition for one day. After incubation, medium was changed conditioned medium (Mϕ 30%) containing 30% (*v/v*) of filtered supernatant of LPS-stimulated RAW264.7 cells for two days. In contrast to conditioned media (Mϕ 30%), control media used a supernatant of non-activated RAW264.7 cells (without LPS activation). For the preparation of conditioned media, cell viability tests of various percentage (10–100%) of supernatant was performed and we confirmed a concentration of conditioned media up to 30%. For measurement of GLP-1 levels, DMEM medium was exchanged with low-glucose (1 g/L) DMEM overnight after differentiation for three days. Following starvation with low-glucose DMEM, differentiated NCI-H716 cells were washed with PBS and 10% (*w/v*) glucose in PBS containing 1 mM CaCl_2_ was treated for 1 h. The Sglt1 inhibitor phloridzin was treated at 100 μM with 10% glucose solution in differentiated NCI-H716 cells.

### 4.8. Isolation Intestinal Epithelium Cells

The small intestine was collected into ice-cold Leibocitz-15 (L-15) medium, and then the collected small intestine was opened, rinsed in L-15, and chopped. The minced tissue was digested five times with 0.35 mg/mL collagenase XI (Sigma) in Dulbecco’s modified Eagle’s medium (4.5 g/L glucose) at 37 °C for 10 min with intermittent vigorous shaking, and the supernatants from digests 3–5 were subjected to centrifuge at 1000× *g* for 3 min. Digested pellets re-suspended in complete DMEM containing 10% FBS (Lonza, Basel, Swiss) and 1% penicillin-streptomycin and filtered by 100 μm strainer. Isolated intestinal epithelium cells were placed on matrigel-coated plates for immunofluorescence and immunoblotting. Cells were incubated for three days and then the culture medium was changed to conditioned media containing a supernatant at 30% of LPS-stimulated RAW264.7 cells for two days.

### 4.9. Immunoblotting

Immunoblotting was performed as preciously described [48]. Tissue samples of the intestine were washed and lysed in T-per buffer (Cell Signaling technology, Danvers, MA, USA) for an hour at 4 °C. Samples were subjected to 8–10% SDS-gel electrophoresis with same amount of 20–30 μg [49]. Membranes were blocked for 5% BSA in TBST and incubated with primary antibody individually overnight.

### 4.10. Statistical Analysis

Data were represented with the means with SEM. The statistical analysis and graphics were performed using GraphPad Prism 5 software package (GraphPad Software, San Diego, CA, USA). The results of blood sampling in mice were analyzed by the two-way ANOVA. Generalized estimating equation for repeated-measures was used to analyze the OGTT, gastrointestinal hormones levels and body weight change to detect group-by-time interactions [50]. An unpaired *t* test was applied for comparison with control in respect of hormone levels and gene expression of each intestine parts, respectively. *P* values * *p* < 0.05, ** *p* < 0.01, *** *p* < 0.001 are considered statistically significant.

## Figures and Tables

**Figure 1 ijms-20-02537-f001:**
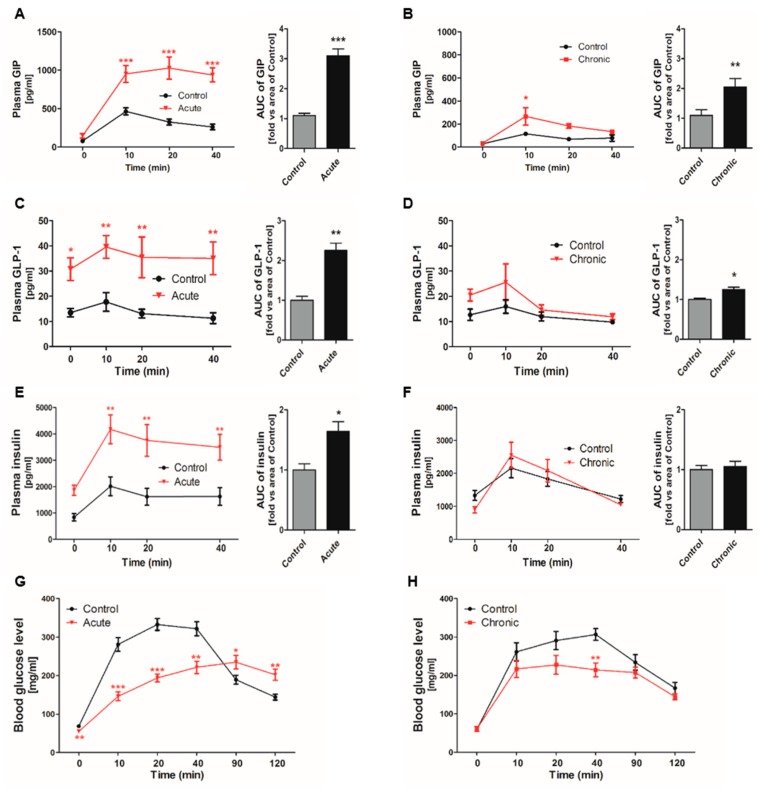
Incretin secretion increased by glucose gavage in gastrointestinal (GI)-inflamed mice. The hormone levels were measured at each time point. Total GIP secretion in response to oral glucose gavage (5 mg/kg) in the acute (**A**) and the chronic GI-inflamed group (**B**). Glucose-induced GLP-1 secretion in the acute (**C**) and chronic inflamed groups (**D**). Insulin secretion in the acute (**E**) and chronic inflamed groups (**F**). The plasma glucose concentrations were measured by OGTT under starvation conditions in the acute (**G**) and chronic groups (**H**). Black line: Control, red line: GI-inflamed mice, *n* = 5/group, * *p* < 0.05, ** *p* < 0.01, *** *p* < 0.001, one-way ANOVA. Grey bar: Control; black bar: GI-inflamed mice groups. * *p* < 0.05, ** *p* < 0.01, *** *p* < 0.001, Student’s *t*-test.

**Figure 2 ijms-20-02537-f002:**
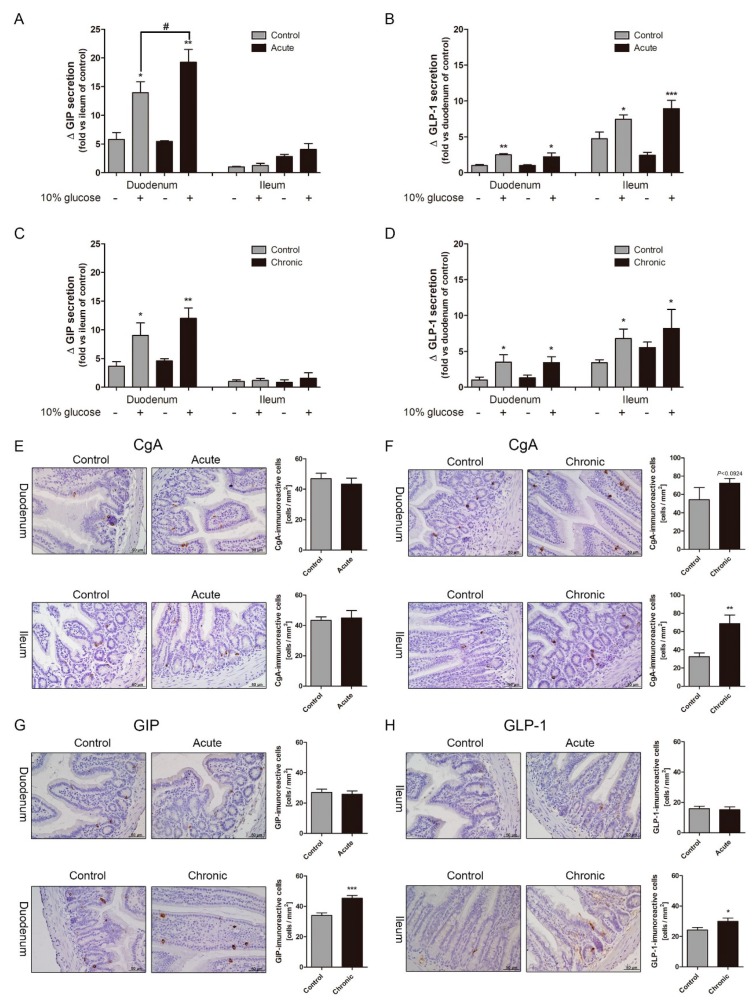
Gastrointestinal tract (GI) inflammatory conditions stimulate glucose-induced incretin secretion. Glucose-stimulated GIP secretion (**A**) and GLP-1 secretion (**B**) in the minced duodenum and ileum tissues of the acute group. Incretin secretion in response to 10% glucose in the chronic group (**C**,**D**). Comparison of the absence and presence of 10% glucose. * *p* < 0.05, ** *p* < 0.01, *** *p* < 0.001, *n* = 6/group, Compared to the control group in presence of 10% glucose. ^#^
*p* < 0.05, Student’s *t*-test. The distribution of Chromogranin A (CgA), a representative enteroendocrine cell marker, in the acute (**E**) and chronic groups (**F**). GIP-positive spots in duodenum tissue (**G**) and GLP-1-positive spots in ileum tissue (**H**). Positive cells (brown spots). Graph shows mean ± SEM. * *p* < 0.05, ** *p* < 0.01, *** *p* < 0.001, Student’s *t*-test. Scale bars represent 50 μm.

**Figure 3 ijms-20-02537-f003:**
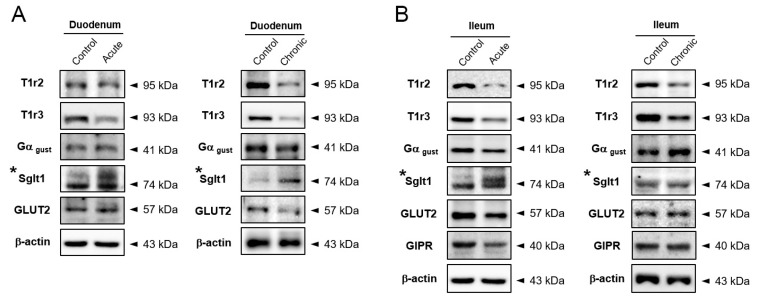
Na+-glucose co-transporter Sglt1 was increased in GI-inflamed mice. The protein expression levels of glucose-sensing receptors including subunits of sweet taste receptor, Sglt1, and GLUT2 in duodenum (**A**) and ileum tissue (**B**) of GI-inflamed mice.

**Figure 4 ijms-20-02537-f004:**
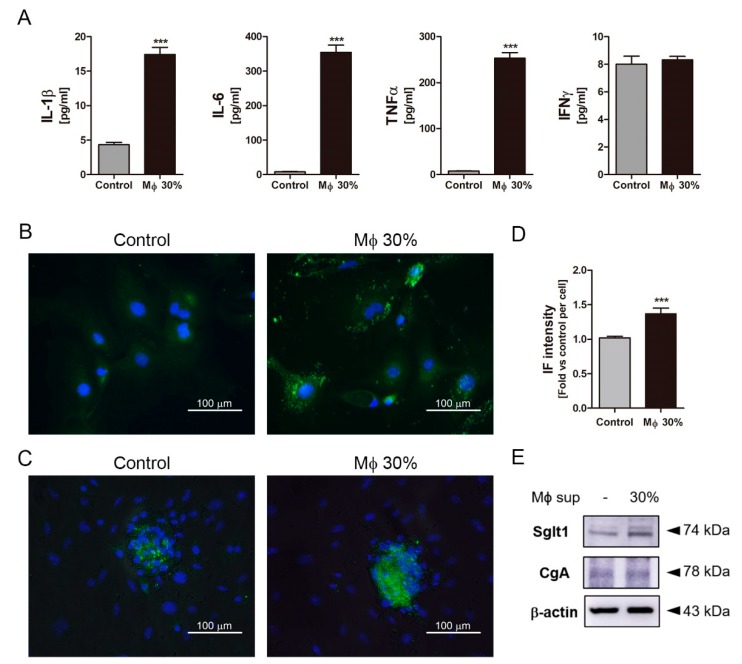
The inflammatory condition induced an increment of Na+-glucose co-transporter Sglt1 in isolated intestinal epithelium cells. Conditioned media (Mϕ 30%) containing inflammatory cytokines, such as IL-1β, IL-6, TNFα, and IFNγ (**A**). IFNγ did not show significant difference. Isolated intestinal epithelium cells (IECs) were incubated without Mϕ 30% (Control, using a supernatant of non-activated RAW264.7 cells) or with conditioned media (Mϕ 30%) for two days. Following incubation with conditioned media (Mϕ 30%) for two days, Sglt1 was increased in single cells (**B**) and crypt particles (**C**). Scale bars represent 100 μm. Intensity of Sglt1 fluorescence elevated with incubation of Mϕ 30% (**D**). The protein expression levels of Sglt1 and Chromogranin A (CgA) in isolated IECs (**E**). *** *p* < 0.001, *n* = 5–6/group, Student’s *t*-test.

**Figure 5 ijms-20-02537-f005:**
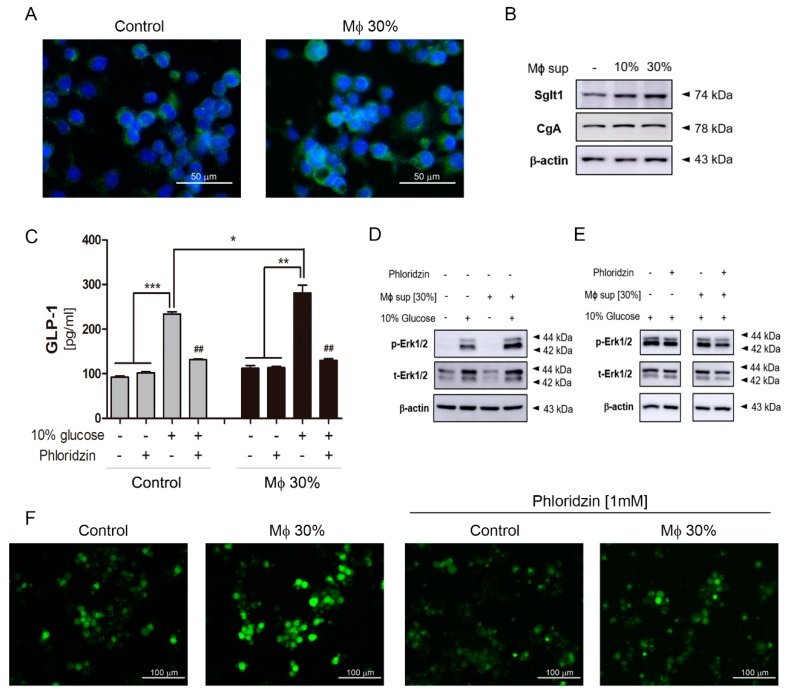
The inflammatory condition induced an increment of Na+-glucose co-transporter Sglt1 in differentiated NCI-H716 cells. Fluorescence of Sglt1 in differentiated NCI-H716 cells (**A**). Scale bars represent 50 μm. Sglt1 increased after incubation with conditioned media (Mϕ 30%) in a dose-dependent manner (**B**). GLP-1 secretion in response to 10% glucose in NCI-H716 cells (**C**). Gray bar: Control group; black bar: incubated group with conditioned media. Glucose exposure induces ERK phosphorylation (**D**) and ERK phosphorylation is slightly reduced by incubation with phloridzin (**E**). * *p* < 0.05, ** *p* < 0.01, *** *p* < 0.001. Student’s *t*-test. Images of Fura-4AM fluorescence in differentiated NCI-H716 cells after incubation with 10% glucose and showing effect of 1 mM Phloridzin (**F**). Scale bars represent 100 μm.

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
