# Peer review of "GI inflammation Increases Sodium-Glucose Cotransporter Sglt1"

_ijms, 2019, doi:10.3390/ijms20102537_

Round 1

Reviewer 1 Report

The revised manuscript is well answered my concerns. Thus, I think this manuscript is worth publishing journal.

Academic Editor Notes

The authors demonstrated a relationship between GI inflammation and glucose metabolism (SGLT1 and incretin) in vivo and in vitro. This provides useful information. There are some problems in the current contents.

1. p4, line 5: The authors described GIP secretion by glucose stimulation was increased in the chronic inflammatory group more than that in the control group. However, in Fig 2c, there was no difference between in the control group and in the chronic inflammatory group (p = 0.1975).

2. Fig 2: Concerning GIP, glucose-stimulated GIP secretion was significantly increased in the acute inflammatory group more than that in the control group in duodenal tissue, but there was no difference of GIP positive cells in duodenal tissue. Glucose-stimulated GIP secretion was not different between in the chronic inflammatory group and in the control group in duodenal tissue, but GIP positive cells in the chronic inflammatory group were more than that in the control group in duodenal group. The authors should mention the reason in more detail.

The causes of elevated glucose-induced hormone secretion are an increase of hormone production and reactivity about stimuli. In our results, we confirmed over-secretion of incretin hormone in response to glucose exposure in the acute inflammatory group more than that in the chronic group. The acute inflammatory group (severe inflammation) showed a significant increase of Sglt1 protein, whereas, the chronic group showed a slightly increase of hormone production and Sglt1 protein in duodenum tissue. Over-secretion of incretin hormone was due to the elevated Sglt1 protein in the acute group, whereas the chronic group was affected by the increase of hormone production rather than the increase of Sglt1 protein. The effect of Sglt1 is greater than that of hormone production.

In Figure 2C, we filled out p value (p=0.1975). The notation seems to have caused confusion. We revised and described these mention in Discussion.

Ex vivo, incretin hormone secretion by glucose exposure significantly increased without inflammatory cytokines in the acute inflammatory group. In both GI-inflamed group, incretin secretion was higher than the intact control group, but there was not significant compared with in the intact control group except acutely GI-inflamed duodenum tissue.

3. Fig 3: Concerning Sglt1 (slc5a1), Sglt1 protein was increased in the acute inflammatory group more than in the control group in both duodenum and ileum, but Sglt1 mRNA was increased in the acute inflammatory group only in duodenum. Sglt1 protein was increased in the chronic inflammatory group both in duodenum and ileum, but Sglt1 mRNA was not increased in the chronic inflammatory group both in duodenum and ileum. The authors should mention the reason.

The expression of mRNA does not represent all protein production.

Interestingly, slc5a1 gene expression was sharply rose in stomach of the acute group. These mRNA expression data was derived from mouse intestinal tissue, and an increase of mRNA expression can be affected by various factors.

We decided to remove these results because slc5a1 gene expression can be controversial compared with data of protein expression.

4. p7, line 14: not Figure 5E but Figure 5F

We revised in manuscript

5. Fig 5c: What comparison was ##?

When 10% glucose was treated in NCI-H716 cells, without phloridzin group versus with phloridzin group. We added this information in figure legend.

6. Fig 5c: What was black bar and gray bar?

Gray bar: Control group, Black bar: incubated group with conditioned media.

7. I think the conclusion of this study is as follows: GI inflammation - SGLT1 increase - intracellular glucose increase - incretin secretion increase - blood glucose decrease. If so, please add this schema in the manuscript, and mention the mechanism of SGLT1 increase by inflammation.

We additionally mentioned this schema in the manuscript

These results showed enhanced incretin secretion in response to glucose through an increase of Sglt1 protein induced by GI inflammatory conditions, not fasting. Our results show a link between GI inflammation and glucose-dependent incretin secretion. Severe inflammatory condition can lead to more Sglt1 increase than mild inflammation.

Reviewer 2 Report

Unfortunately, the authors didn’t adequately addressed my previous concern. I don’t see any major changes in the manuscript in response to my concerns. The responses in the rebuttal is also very poor, no logical argument. Some major concerns that were not addressed properly are mentioned below.

1.     There is no data or discussion to address the concern why Sglt1  and glut2 were not induced in the colon during acute colitis. The expression of Sglt1 and Glut2 looks similar in all parts of the intestine in control mice (Fig 3C and D). If inflammation induces these genes, why these receptors are not induced in the colon during DSS colitis? what the 2 sets of western blots in Fig 3A and 3B represent? 

2.     I don’t see any logical argument or changes in response to my previous comment#3. The data showing GLP-1 and GIP positive immunoreactions ( Sup fig 3) were removed.  The authors were not responsive to my concerns.

3.     Similarly, my concern why the authors had to measure ERK activation, but not other signaling

pathways, was not addressed? What is the connection of ERK with Sglt1? Why there is no effect

of 30% sup?

4.     Overall, the presentation of the manuscript is poor.

Academic Editor Notes

The authors demonstrated a relationship between GI inflammation and glucose metabolism (SGLT1 and incretin) in vivo and in vitro. This provides useful information. There are some problems in the current contents.

1. p4, line 5: The authors described GIP secretion by glucose stimulation was increased in the chronic inflammatory group more than that in the control group. However, in Fig 2c, there was no difference between in the control group and in the chronic inflammatory group (p = 0.1975).

2. Fig 2: Concerning GIP, glucose-stimulated GIP secretion was significantly increased in the acute inflammatory group more than that in the control group in duodenal tissue, but there was no difference of GIP positive cells in duodenal tissue. Glucose-stimulated GIP secretion was not different between in the chronic inflammatory group and in the control group in duodenal tissue, but GIP positive cells in the chronic inflammatory group were more than that in the control group in duodenal group. The authors should mention the reason in more detail.

The causes of elevated glucose-induced hormone secretion are an increase of hormone production and reactivity about stimuli. In our results, we confirmed over-secretion of incretin hormone in response to glucose exposure in the acute inflammatory group more than that in the chronic group. The acute inflammatory group (severe inflammation) showed a significant increase of Sglt1 protein, whereas, the chronic group showed a slightly increase of hormone production and Sglt1 protein in duodenum tissue. Over-secretion of incretin hormone was due to the elevated Sglt1 protein in the acute group, whereas the chronic group was affected by the increase of hormone production rather than the increase of Sglt1 protein. The effect of Sglt1 is greater than that of hormone production.

In Figure 2C, we filled out p value (p=0.1975). The notation seems to have caused confusion. We revised and described these mention in Discussion.

Ex vivo, incretin hormone secretion by glucose exposure significantly increased without inflammatory cytokines in the acute inflammatory group. In both GI-inflamed group, incretin secretion was higher than the intact control group, but there was not significant compared with in the intact control group except acutely GI-inflamed duodenum tissue.

3. Fig 3: Concerning Sglt1 (slc5a1), Sglt1 protein was increased in the acute inflammatory group more than in the control group in both duodenum and ileum, but Sglt1 mRNA was increased in the acute inflammatory group only in duodenum. Sglt1 protein was increased in the chronic inflammatory group both in duodenum and ileum, but Sglt1 mRNA was not increased in the chronic inflammatory group both in duodenum and ileum. The authors should mention the reason.

The expression of mRNA does not represent all protein production.

Interestingly, slc5a1 gene expression was sharply rose in stomach of the acute group. These mRNA expression data was derived from mouse intestinal tissue, and an increase of mRNA expression can be affected by various factors.

We decided to remove these results because slc5a1 gene expression can be controversial compared with data of protein expression.

4. p7, line 14: not Figure 5E but Figure 5F

We revised in manuscript

5. Fig 5c: What comparison was ##?

When 10% glucose was treated in NCI-H716 cells, without phloridzin group versus with phloridzin group. We added this information in figure legend.

6. Fig 5c: What was black bar and gray bar?

Gray bar: Control group, Black bar: incubated group with conditioned media.

7. I think the conclusion of this study is as follows: GI inflammation - SGLT1 increase - intracellular glucose increase - incretin secretion increase - blood glucose decrease. If so, please add this schema in the manuscript, and mention the mechanism of SGLT1 increase by inflammation.

We additionally mentioned this schema in the manuscript

These results showed enhanced incretin secretion in response to glucose through an increase of Sglt1 protein induced by GI inflammatory conditions, not fasting. Our results show a link between GI inflammation and glucose-dependent incretin secretion. Severe inflammatory condition can lead to more Sglt1 increase than mild inflammation.

Round 2

Reviewer 2 Report

I understand, it might be difficult to address all of the concerns at this stage. However, with the changes the made in revised version, the manuscript is acceptable.

Author Response

# Reviewer Comment

I understand, it might be difficult to address all of the concerns at this stage. However, with the changes the made in revised version, the manuscript is acceptable.

I'm very pleasured with final reviewer comment.  Thank you for giving an oppertunity to be able to revise manuscripts. After this revision, I will investigate cause about the alteration of gut hormone secretion and insulin secretion.

We revised the manuscript. Please check re-submitted version (r3 version).

Again, thank you for your review. 

This manuscript is a resubmission of an earlier submission. The following is a list of the peer review reports and author responses from that submission.

Round 1

Reviewer 1 Report

The manuscript entitled as ‘GI inflammation increases Sodium-glucose cotransporter Sglt1’ by Park, et al. examined the glucose-induced incretin secretion in an inflamed state, and suggested that GI inflammatory conditions induced over-stimulated incretin secretion via an increase of Sglt1. However the paper is lacking data for drawing the conclusion.

Majour comment

1. Figure 2 showed GIP and GLP-1 secretion in the acutely and chronic GI inflamed group after 10% glucose exposure. However, the results of comparison among GI inflamed group and control group are very important and necessary. Therefore, the authors should please add these data in figure 2.

2. The authors showed in Figure 2E, 2F and Supplementary Figure 3 that chronic inflammation increases amount of secretory vesicle in small intestine. The authors stated that these vesicle contained GIP and GLP-1. However, there is no direct evidence which shows the vesicle contains GIP and GLP-1. The authors need to show the data.

3. Figure 2 showed that the over-secretion of incretin was induced in response to glucose exposure in chronic GI-inflamed state. However, the protein and mRNA associated with Sglt1 were changed in acute GI-inflamed mice (Figure 3). Authors should discussed about these difference.

4. Authors showed the results of GLP-1 secretion in differentiated NCI-H716 cells in Figure 5C. The authors should show the result of GIP secretion at the same experiment, because GIP secretion also important.

5. In Figure 5C, the results of GLP-1 and GIP secretion after incubation with condition media (Macrophage 30%) were interested. Therefore, the authors are suggested to show these data.

Reviewer 2 Report

In this study, the authors investigated the role of inflammatory responses in glucose homeostasis focusing on the role of SgLt1 in the regulation of gut hormone GLP1 and GIP. It has been shown that, during acute DSS-induced colitis, plasma levels of GIP, GLP1, and insulin are increased, particularly in response to glucose. Ex vivo colon culture was used to show that GLP1 and GIP are induced by intestinal epithelial cells in response to glucose. Further, they showed that these hormones are regulated by Sglt1, which is a receptor for intestinal glucose and induced by inflammatory responses in the gut. Overall, the authors have performed a series of experiments  investigating how glucose homeostasis in the gut is maintained through the regulation of hormones involved in glucose metabolism. Overall the study is interesting. However, there are some concern regarding clarity and inconsistency of the data and their interpretation.

1.     In describing the figure 1, it is mentioned in the text that mice was challenged with 10% glucose. While figure legend stated that mice were orally gavaged with 5mg/kg glucose. Which is correct? Also, the methods of these experiments should be clearly described in the Methods section. and adequately in the text or figure legends. How the hormones were measured after glucose challenge. Is the serum was collected from same mice at different time, or different batch of mice was sacrificed at each time point? Experimental procedure should be described clearly.

2.     Colitis model was used to show the impact of inflammatory responses in the expression of Sglt1 and other gut hormones. DSS-colitis mainly affects colon but not the small intestine. A major criticism is that the authors showed that there is no change in these molecules in the colon. Only change was seen in the duodenum (Figure 2 and 3). The authors should clarify this issue. Is there increased inflammatory response in the duodenum following DSS administration? Why gut hormones and Sglt1 are not increased in the colon, which is the most inflamed area during DSS treatment.

3.     The authors argued that GI-inflammation  increases gut hormone secretion (page 4, line 126-128). However, Supplementary Figure 3 doesn’t support this conclusion. There was no increase in mRNA levels of gip, gcg, and pcsk1 in the intestine during acute colitis (Sup Fig 3C), whereas plasma levels of these hormones were highly increased in acute colitic mice (Figure 1). Ex vivo colon explant study also showed that intestine from acute colitis expressed higher GIP and GLP1 (Figure 2). These data are inconsistent.

4.     The rational of measuring p-ERK in Figure 5D is not clear. The data shown here is very confusing and not matching with what the authors described in the text. There was no increase of p-ERK when incubation with 30% Sup. Only effect is seen with 10% glucose.

5.     What two different bars, gray and black, mean in figure 5C? Phloridzin treatment lowers the expression of GLP1 to almost basal level (Figure 5C), whereas it minimally affect p-ERK (Figure D). Is there any explanation?

6.     Why intracellular Ca2+ was measured in Figure 5E.  How this data is connected with Sglt1 and gut hormones? There is a lack of describing proper rationale of major experiments.

7.     Supplementary figures 1, 4, and 5 were not mentioned and discussed in the result sections.  Supplementary figure 3 and 4 were also not adequately discussed in the result sections.

8. There are some language issues which must be corrected with English editing.